# Fish predators control outbreaks of Crown-of-Thorns Starfish

Frederieke J. Kroon [1✉], Diego R. Barneche [2,3] & Michael J. Emslie [1]

Outbreaks of corallivorous Crown-of-Thorns Starfish (CoTS, *Acanthaster* spp.) have caused persistent and widespread loss of coral cover across Indo-Pacific coral reefs. The potential drivers of these outbreaks have been debated for more than 50 years, hindering effective management to limit their destructive impacts. Here, we show that fish biomass removal through commercial and recreational fisheries may be a major driver of CoTS population outbreaks. CoTS densities increase systematically with increasing fish biomass removal, including for known CoTS predators. Moreover, the biomass of fish species and families that influence CoTS densities are 1.4 to 2.1-fold higher on reefs within no-take marine reserves, while CoTS densities are 2.8-fold higher on reefs that are open to fishing, indicating the applicability of fisheries-based management to prevent CoTS outbreaks. Designing targeted fisheries management with consideration of CoTS population dynamics may offer a tangible and promising contribution to effectively reduce the detrimental impacts of CoTS outbreaks across the Indo-Pacific.

[1] Australian Institute of Marine Science, Townsville, QLD 4810, Australia. [2] Australian Institute of Marine Science, Crawley, WA 6009, Australia. [3] Oceans Institute, The University of Western Australia, Crawley, WA 6009, Australia. ✉email: f.kroon@aims.gov.au

Predation by native Crown-of-Thorns Starfish (CoTS, *Acanthaster* spp.) during periodic population outbreaks is a major contributor to sustained declines in coral cover across Indo-Pacific coral reefs[1–3]. Phenomenal fluctuations in CoTS abundance within periods of a few years[2] can result in increased CoTS densities that cause extensive coral mortality[4], suppressing coral recruitment and recovery[5], and fundamentally altering the form and structure of coral reefs and their biological communities[6]. Destructive impacts of CoTS outbreaks on Indo-Pacific coral reefs were first documented in the 1950s[7] and 1960s[8,9], though anecdotal reports suggest that CoTS population densities have long varied widely[10]. Whether human interference has exacerbated CoTS outbreaks, through increasing nutrient levels enhancing larval recruitment[11] or harvesting of natural predators that would limit CoTS abundance[9], remains a contentious topic[2]. While neither hypothesis has received universal or unequivocal support[2], their contributions to causing or propagating CoTS outbreaks are also not mutually exclusive. Given that CoTS outbreaks continue to be one of the major drivers of coral loss[12], including during recent mass bleaching events[13], new pathways for CoTS control at large scale have become increasingly important to halt further declines in coral cover and support reef restoration and resilience in a warming climate across the Indo-Pacific[14].

Contemporary management to reduce the detrimental impact of CoTS population outbreaks on coral reefs centres around a combination of direct manual control and water quality improvement[12,15]. Manual control programmes have been implemented since the 1960s[7,16], killing or removing an estimated 17 million CoTS across the Indo-Pacific by 2014[2], but with limited success in reducing either CoTS densities or coral loss[17]. Recent improvements in single-injection methods to cull individual CoTS[18], combined with strategic deployment of manual control[19], have significantly improved the efficacy of manual control programmes to reduce CoTS impacts on coral reefs[14,15]. The role of water quality improvement programmes in CoTS control are predicated on the hypothesized link between nutrient enrichment and CoTS outbreaks[11]. The nutrient enrichment hypothesis states that high nutrient availability increases phytoplankton biomass and enhances CoTS larval growth and survival leading to mass recruitment events and outbreaks. In Australia, the purported role of nutrient enrichment from land-based run-off (i.e. the terrestrial run-off hypothesis)[11] has become a central argument for policy and investment to improve water quality of the Great Barrier Reef (GBR) World Heritage Area[20]. While its efficacy is supported by independent modelling exercises[14,21], catchment and water quality improvement efforts, implemented since the early 2000s[22], are unlikely to have acted to suppress CoTS population dynamics and cannot yet be relied upon to contribute to CoTS control on the GBR[15].

The predator removal hypothesis, first put forward in the 1960s, posits that release from predation pressure is a primary cause of CoTS outbreaks[9]. In contrast to water quality improvement, management of predators has never been a component of controlling CoTS population outbreaks. Initially, a decrease in population densities of the giant triton (*Charonia tritonis*) was thought responsible, being the only known predator of adult CoTS[9]. Since then, close to 100 species of coral reef organisms have been reported to consume pelagic larvae and benthic juvenile, sub-adult and adult CoTS[23], including 80 coral reef fish species from 17 families (Table 1)[24]. Recently, CoTS DNA was detected in faecal and gut content samples from 18 wild-caught coral reef fish species, including nine fish species that had not previously been reported to feed on CoTS[24]. This further indicates that more coral reef fish species might predate on pelagic and benthic CoTS than is currently appreciated. Potential

regulation of CoTS outbreaks by coral reef fish is most likely to occur through predation on the pelagic phase controlling the settlement of CoTS larvae, and (sub-) lethal predation on the benthic phases, influencing growth, reproduction[25] and mortality, in particular during low, non-outbreak, densities[26,27]. While independent modelling studies have provided support for coral reef fish predation regulating CoTS outbreaks[26–29], the lack of data on harvesting of natural predators, particularly at spatio-temporal scales large enough to encompass CoTS outbreak dynamics (but see ref. 45 for a smaller-scale example), means we still have limited understanding of how removal of coral reef fish may affect CoTS abundance and the applicability of predator-based management to prevent CoTS outbreaks.

In this study, we examine whether recent population outbreaks of the Pacific Crown-of-Thorn Starfish (*Acanthaster* cf. *solaris*) in the GBR Marine Park are influenced by removal of coral reef fish through commercial and recreational fisheries, which have been the major extractive activities on the GBR since at least the 1950s (Supplementary Method 1)[30,31]. Our study combines CoTS and coral reef fish monitoring data and fisheries retained catch data independently collected across the 344,400 km$^2$ GBR Marine Park since 1985 to (1) determine whether CoTS densities can be predicted by the removal of coral reef fish biomass; (2) quantify the effects of no-take marine reserves (Supplementary Table 1) on the biomass, density and size of fish groups that influence CoTS densities; and (3) evaluate whether fisheries management is applicable to CoTS control by comparing CoTS densities between reefs within no-take marine reserves to those that are open to fishing. We demonstrate that (1) CoTS densities increase systematically with increasing fish biomass removal, including for known CoTS predators; (2) the biomass of fish species and families that influence CoTS densities are 1.4–2.1-fold higher on reefs within no-take marine reserves, and (3) CoTS densities are 2.8-fold higher on reefs that are open to fishing. Our results strongly suggest that fish biomass removal through commercial and recreational fisheries may be a major driver of CoTS population outbreaks, indicating the applicability of targeted fisheries-based management to prevent CoTS outbreaks.

## Results

**Response of CoTS density to fish biomass removal.** Coral reef fisheries may affect CoTS density through the removal of fish species that either directly predate on[24] or indirectly influence predation on CoTS[21]. Here we focussed on six fish groups based on their reported consumption of benthic CoTS (Table 1)[24] and their contribution to the commercial and recreational charter fisheries in the GBR Marine Park (Supplementary Table 2; Supplementary Data 1)[32]. These six groups comprised of (1) Labridae (wrasses), (2) Lethrinidae (emperors), (3) *Lethrinus miniatus* and *L. nebulosus* (redthroat and spangled emperors), (4) Lutjanidae (tropical snappers), (5) Serranidae (rockcods) and (6) *Plectropomus* spp. and *Variola* spp. (coral trout). Potential effects of other known predators of pelagic and benthic CoTS, including species of the families Ballistidae (triggerfish), Chaetodontidae (butterflyfish), Diodontidae (Porcupinefish), Haemulidae (grunters), Pomacentridae (damselfish) and Tetraodontidae (Pufferfish) (Table 1)[24], could not be assessed as data on fish biomass removal by fisheries for these groups were either not available or too limited for our analyses (Haemulidae).

We employed a Bayesian hurdle-gamma modelling approach using paired fisheries retained catch and CoTS density observations encompassing three CoTS outbreak cycles (19,239 paired observations for 157 individual fisheries logbook reporting sites between 1990 and 2018, see "Methods"; Supplementary Methods 1 and 2 and Supplementary Fig. 1). Because fish removal effects on

**Table 1 Coral reef fish known to consume Crown-of-Thorns Starfish and targeted by fisheries in the Great Barrier Reef Marine Park.**

| Coral reef fish family | | Consumption of CoTS | | | | Coral Reef Fin Fish Fisheries | | | |
|---|---|---|---|---|---|---|---|---|---|
| Scientific name | Common name | Pel. | Ben. | Gut | DNA | Comm. (%) | Chart. (%) | Rec. (%) | Ind. (%) |
| Apogonidae | Cardinalfish | | | | Y | | | | |
| Ballistidae | Triggerfish | | Y | Y | Y | | | | |
| Chaetodontidae | Butterflyfish | Y | | | | | | | |
| Diodontidae | Porcupinefish | | | Y | | | | | |
| Gobiidae | Gobies | | Y[c] | | | | | | |
| Haemulidae | Sweetlips | | | | Y | Y (0.3) | Y (0.6) | Y | |
| Holocentridae | Soldierfish | | | | Y | | | | |
| Labridae[a] | Wrasses | | Y[c] | Y | Y | Y (0.4) | Y (5.2) | | |
| | Parrotfish[b] | | Y[c] | | | | | | |
| Lethrinidae[a] | Emperors | | Y | Y | Y | Y (12.9) | Y (28.8) | Y | |
| Lutjanidae[a] | Tropical snappers | | Y[c] | | Y | Y (14.0) | Y (16.7) | Y | Y |
| Mulllidae | Goatfish | | Y[c] | | | | | | |
| Nemipteridae | Monocle bream | | Y[c] | | | | | | |
| Pomacanthidae | Angelfish | | Y[c] | | | | | | |
| Pomacentridae | Damselfish | Y | Y[c] | | Y | | | | |
| Serranidae[a] | Rockcods | | | Y | Y | Y (45.7) | Y (25.1) | Y | Y |
| Tetraodontidae | Pufferfish | | Y | | Y | | | | |

Consumption of pelagic (Pel.) and benthic (Ben.) Crown-of-Thorns Starfish (CoTS, *Acanthaster* spp.) has been observed directly, inferred from gut content analyses (Gut), or from DNA analyses of faecal or gut content samples (DNA)[24]. Coral reef fish families that are most commonly targeted by Coral Reef Fin Fish Fisheries, including commercial line (Comm.), recreational charter (Chart.), recreational (Rec.) and Indigenous (Ind.) fisheries, in the Great Barrier Reef Marine Park, Australia, based on information from Queensland Department of Agriculture and Fisheries, are presented. Proportional contribution of each family to the total coral reef fish biomass harvested by commercial line and recreational charter fisheries are given for 2018 (Supplementary Data 1). Note that harvests across coral reef fish families and fisheries vary considerably (Supplementary Data 1–3).
[a]Coral reef fish families monitored by the Australian Institute of Marine Science's Long-Term Monitoring Programme (see Supplementary Table 5 for species monitored within the Labridae, Lethrinidae, Lutjanidae and Serranidae families and considered in this study).
[b]Parrotfish were previously considered a separate family (Scaridae) but are now placed as a sub-family Scarinae within the Labridae.
[c]Fish species within these families observed consuming injured, moribund and/or dead CoTS only.

CoTS density might be manifested after multiple time lags[28], we ran six models (i.e. annual time lags between fish biomass removal and CoTS density of 1–6 years) for each of these six groups. CoTS densities increased with increasing biomass removal of coral reef fish (Fig. 1, Supplementary Figs. 2 and 3 and Supplementary Table 3), most substantially (>95% probability) with increasing retained catches of redthroat and spangled emperors (*L. miniatus* and *L. nebulosus*) and of the emperor family (Lethrinidae). This family, and particularly the two species, are among the most frequently and well-documented predators of benthic CoTS (Table 1)[24]. Positive effects of fish biomass removed by fisheries on CoTS densities were also detected (both >80 and >95% probabilities) for tropical snappers (Lutjanidae) and rockcods (Serranidae). Interestingly, we found similar effects for coral trout (*Plectropomus* spp. and *Variola* spp.)—large piscivores[33] that have not yet been reported to consume CoTS but comprise the primary target species for fisheries in the GBR Marine Park (Supplementary Method 1 and Supplementary Data 1 and 2). Across these five fish groups, the positive effects of biomass removal on CoTS densities were most pronounced (both >80 and >95% probabilities) following time lags of 1, 2 and 4 years but less common following time lags of 3, 5 and 6 years (Fig. 1, Supplementary Fig. 2 and Supplementary Table 3). In contrast, biomass removal of wrasses (Labridae) did not influence CoTS density during any of the six time lags examined.

Our finding that CoTS densities increased with increasing biomass removal of coral trout is unexpected given that adult coral trout are almost entirely piscivorous[33]. This result may be related to levels of coral trout biomass removed corresponding with similar levels of biomass removed in other fish groups. Fish biomass removed did indeed coincide strongly between coral trout and Serranidae (pairwise Pearson correlation values >0.900 for all six time lags) and between redthroat and spangled emperors (*L. miniatus* and *L. nebulosus*) and Lethrinidae (>0.900 for four of the six time lags) (Supplementary

Table 4). This aligns with the commercial line fishery primarily targeting coral trout and redthroat emperor, with another 20 species targeted including emperor, tropical snapper and rockcod species (Supplementary Data 1)[32]. In contrast, fish biomass removed coincided less strongly between coral trout and redthroat and spangled emperors (range of pairwise Pearson correlation values: 0.155–0.650) and between coral trout and Lethrinidae (0.564–0.678) (Supplementary Table 4). Thus, while some interaction between the different fish groups cannot be discounted, the positive effects of coral trout biomass removed by fisheries on CoTS densities is unlikely to be solely due to these interactions. While CoTS consumption has not been reported for coral trout, juvenile *Plectropomus leopardus* feed mainly on benthic invertebrates[34] and further research on whether this may include recently settled or juvenile CoTS is warranted. If juvenile coral trout indeed consume CoTS, removal of adult coral trout may indirectly influence predation on CoTS given the importance of high levels of self-recruitment in maintaining *Plectropomus* populations[33]. Our finding could also support the involvement of more complex ecological effects in modulating CoTS outbreaks[21], which have remained speculative until now. Such effects may include the ecological release of invertebrates that prey on juvenile CoTS when higher numbers of large piscivores are present; removal of such piscivores may result in increased densities of benthic carnivorous fishes[35]. It could also include the behavioural release of planktivorous fishes away from the reef substrate with reduced numbers of large piscivores, resulting in increased number of CoTS larvae able to reach the benthos[36]. The potential involvement of complex ecological effects is further supported by our finding that the effects of *Plectropomus* spp. and *Variola* spp. and of Serranidae removal on CoTS density were most strongly manifested after 4 years (Supplementary Table 3). In contrast, the fish removal effects of well-known CoTS predators, namely, *L. miniatus* and *L. nebulosus*, Lethrinidae and Lutjanidae (Table 1), were most pronounced after 1 and 2 years (Supplementary Table 3).

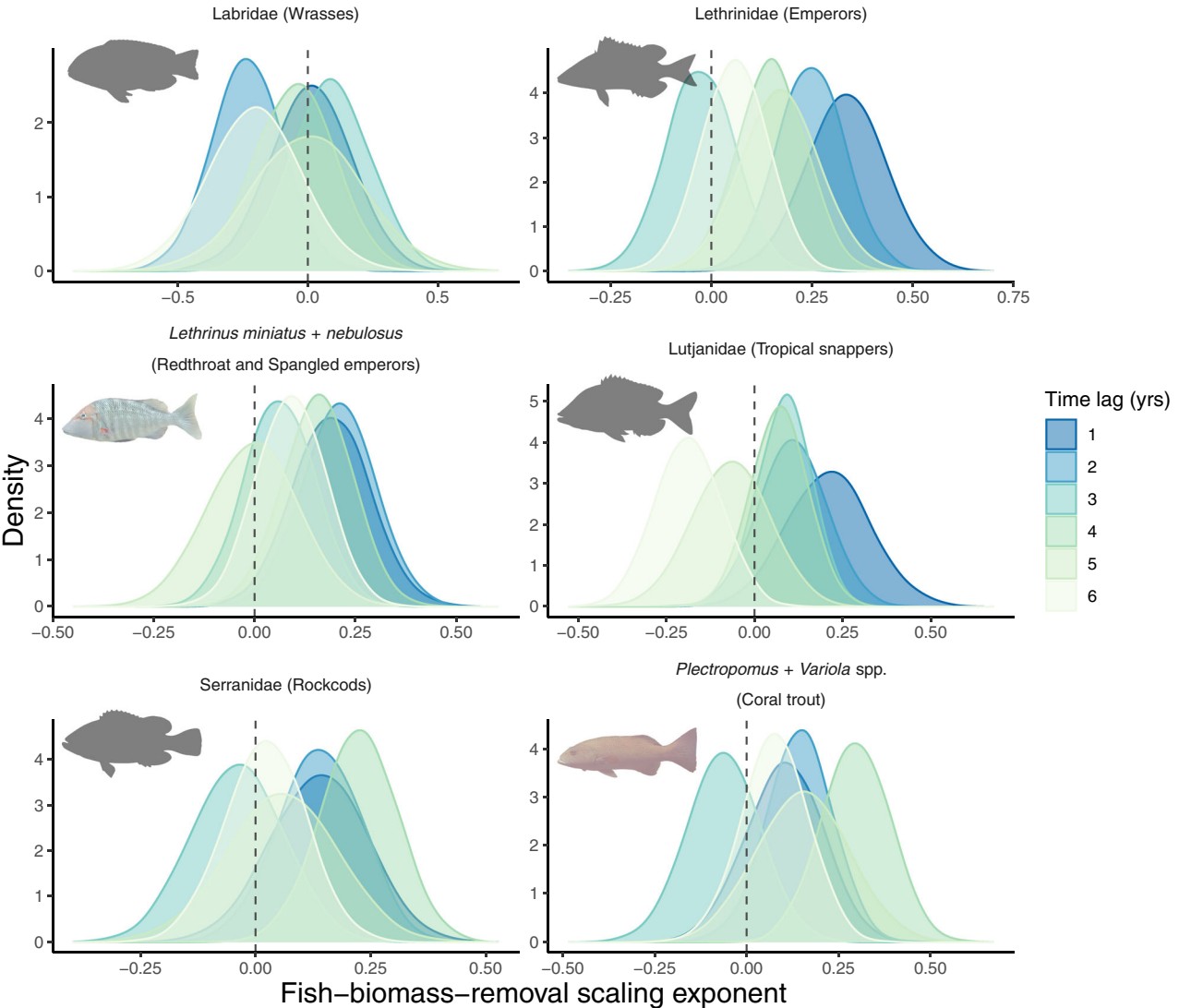

**Fig. 1 Response of Crown-of-Thorns Starfish density to fish biomass removal.** Posterior distributions of the biomass-removal scaling exponent of Pacific Crown-of-Thorns Starfish (CoTS, *Acanthaster* cf. *solaris*) density. For each target fish group and time lag, the exponent was estimated independently using a hurdle-gamma Bayesian model (see "Methods"; Supplementary Method 2). Exponents are equivalent to the fish biomass-removal slope on the linear scale (log) of the gamma (i.e. non-zero) component of the model. Vertical dashed lines represent zero, and values above this line constitute evidence suggesting that fish biomass removal drives increase in CoTS density over time. Source data are provided as a Source data file. Fish silhouettes for Lethrinidae, Serranidae and *L. miniatus* from AIMS ©, for *P. leopardus* from F. Kroon ©, for Lutjanidae from R package fishualize (https://github.com/nschiett/fishualize, License GPL-2) and for Labridae cut and adapted to a silhouette from Fig. 2C (*Choerodon anchorago*, terminal phase male) in Gomon[63].

These arresting findings demonstrate that CoTS densities can be predicted by fish biomass removal, including, but not limited to, known predators of benthic juvenile, sub-adult and adult CoTS. These effects are likely stronger, given that the take of emperors, tropical snappers and rockcods by the recreational (non-charter) fisheries in the GBR Marine Park is estimated to have been of similar magnitude to that of the commercial line fisheries for decades ("Methods"; Supplementary Method 1 and Supplementary Data 1 and 2)[30,37]. Furthermore, the fish removal effects on CoTS density were most strongly manifested within 2 years for *L. miniatus* and *L. nebulosus*, Lethrinidae and Lutjanidae and 4 years for *Plectropomus* spp. and *Variola* spp. and Serranidae. This indicates that targeted fisheries-based management of CoTS may provide an effective approach to rapidly reduce CoTS densities and contribute to preventing outbreaks across the Indo-Pacific.

**Effects of no-take marine reserves on coral reef fish.** Effective no-take marine reserves are known to increase fish biomass and

species richness of large fishes within such reserves[38–40], including for fisheries target species[41]. Here we determine whether these effects also hold for the specific fish groups that influence CoTS densities (Fig. 1). We employed a Bayesian hierarchical model based on fisheries-independent field observations on each of the six fish groups (Supplementary Table 5) to compare biomass, density and length between reefs within no-take marine reserves and reefs that are open to fishing (840 transects conducted biennially on 56 paired fished and unfished reefs from 2006 to 2020, see "Methods"; Supplementary Method 3). Fish biomass was 1.4–2.1-fold higher on unfished reefs compared to fished reefs across all six fish groups (Fig. 2a and Supplementary Table 6), and most likely so (>95% probability) for redthroat and spangled emperors, for coral trout and for the emperor, tropical snapper and rockcod families. Similarly, fish density (Fig. 2b) and fish length (Fig. 2c) were 1.2–1.7-fold higher on unfished reefs (>80% probability) for most fish groups (Supplementary Table 6), and most likely so (>95% probability) for coral trout and for the

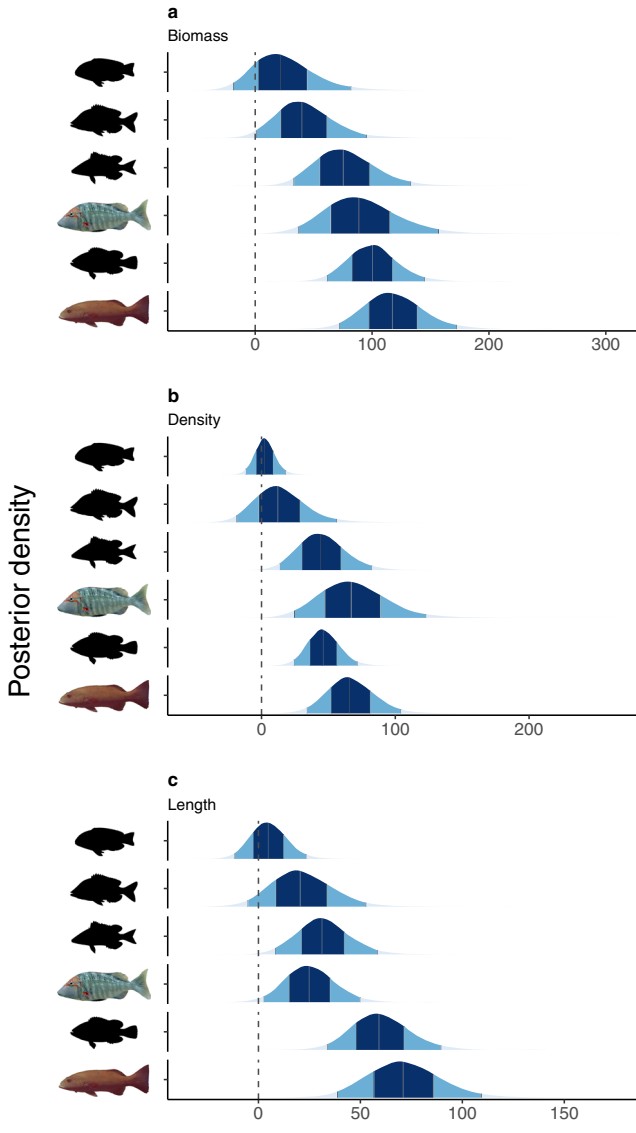

**Fig. 2 Effects of no-take marine reserves on coral reef fish.** Posterior distributions of **a** biomass, **b** density and **c** length, with posterior densities indicating the modelled median differences (middle line) between unfished (U) and fished (F) reefs, with associated 60% (dark blue) and 95% (lighter blue) credible intervals (see "Methods"; Supplementary Method 3). A positive effect indicates higher values on unfished reefs. In each plot, results are presented from top to bottom for (1) Labridae (wrasses), (2) Lutjanidae (tropical snappers), (3) Lethrinidae (emperors), (4) *Lethrinus miniatus* and *L. nebulosus* (redthroat and spangled emperors), (5) Serranidae (rockcods) and (6) *Plectropomus* spp. and *Variola* spp. (coral trout). Source data are provided as a Source data file. Fish silhouettes for Lethrinidae, Serranidae and *L. miniatus* from AIMS ©, for *P. leopardus* from F. Kroon ©, for Lutjanidae from R package fishualize (https://github.com/nschiett/fishualize, License GPL-2) and for Labridae cut and adapted to a silhouette from Fig. 2C (*Choerodon anchorago*, terminal phase male) in Gomon[63].

rockcod family (both density and length) and for redthroat and spangled emperors and for the emperor family (length only).

**Effects of no-take marine reserves and coral cover on CoTS density.** These striking effects of no-take marine reserves

on fish biomass (Fig. 2a), density (Fig. 2b) and length (Fig. 2c) for fish groups that influence CoTS densities within a few years (Fig. 1) portends the applicability of targeted fisheries-based management to prevent CoTS outbreaks. Using fisheries-independent field observations on CoTS encompassing three CoTS outbreak cycles (157,348 manta tows from 490 reefs between 1985 and 2020, see "Methods"; Supplementary Methods 1 and 4) and a Bayesian hierarchical model that accounts for habitat differences, we found that reefs open to fishing have nearly three times more individual CoTS per tow compared to reefs within no-take marine reserves (2.8 [1.9–3.9], median [95% credible intervals (C.I.s)]; Fig. 3a). These effects were consistent despite CoTS density decreasing systematically with coral cover ($\beta_2 = -0.45$; C.I.s: $-0.67$ to $-0.23$; Fig. 3b) and CoTS density varying >1800-fold ($e^{2 \times \sigma_\zeta}$, where $\sigma_\zeta = 3.77$; C.I.s: 3.58–3.97) across reefs and years. Previous studies have alluded to an effect of no-take marine reserves on CoTS outbreaks[15,35,42]. Our study, capturing the longer-term dynamics of several CoTS outbreaks, strongly demonstrates higher CoTS densities on reefs open to fishing and provides, at equivalent spatio-temporal scales, a fisheries-based mechanism (Figs. 1 and 2) for these effects.

**Discussion**

Combined, our results support the hypothesis that fish biomass removal is a major driver of CoTS population outbreaks. Our study demonstrates the effect of predator removal on CoTS, merging two independent data sets spanning three decades and 344,400 km² in spatial coverage that capture time lags between fishing and changes in CoTS densities, nonlinear dynamics of three CoTS outbreaks[2] and the effects of trophic cascades, including predation[21,24]. Removal of fish predators is likely to contribute to CoTS outbreaks across Indo-Pacific coral reefs, with emperors, tropical snappers and rockcods comprising sizeable components of coastal fisheries catches in this region[43,44]. The role of fish predation in regulating CoTS outbreaks has previously been explored and described in various independent modelling studies[26–29] and inferred following a one-off, small-scale comparative study on fishing intensity[45] but not been substantiated with field observations as reported here. Critically, our findings support the importance of hypothesised ecological and behavioural release[35,36], following removal of the piscivorous coral trout, in influencing CoTS population dynamics and outbreaks. The involvement of more complex ecological effects in modulating CoTS outbreaks[21] have remained speculative until now; our results warrant further research to better understand and quantify the potential role of such trophic cascades. Given that fish predation on benthic CoTS is thought to be particularly important in regulating outbreaks at low CoTS densities[26,27], due to Type II or Type III functional responses[45], removal of even small amounts of biomass of key predatory fish may be sufficient to trigger outbreaks. Indeed, small amounts of fish biomass removal are correlated with CoTS trespassing the threshold towards potential outbreaks (Supplementary Table 7), supporting earlier modelling studies on fish predation regulating CoTS outbreaks[26,27]. Fish biomass was also only up to 2.1-fold higher in no-take marine reserves (Fig. 2), with unfished reefs having markedly lower CoTS densities (Fig. 3). Further, emperor densities across Indo-Pacific reefs and regions with and without CoTS outbreaks were about threefold higher on reefs and fivefold higher in regions that had not experienced CoTS outbreaks[27]. The temporal alignment of an increased exploitation of coral reef fisheries in the 1950s[43] combined with the first reports of CoTS outbreaks across the Indo-Pacific region in the 1950s[7] and 1960s[8,9] further supports the premise that recent CoTS outbreaks

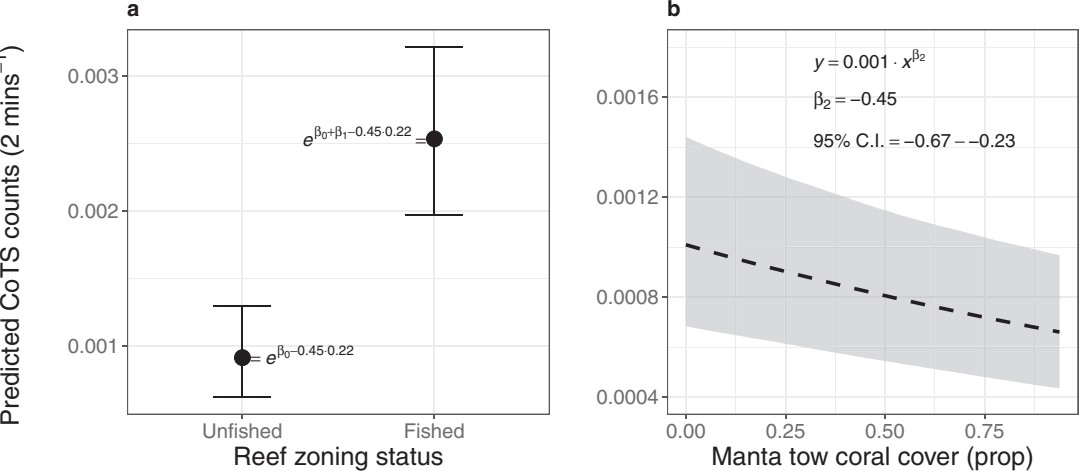

**Fig. 3 Effects of no-take marine reserves and coral cover on Crown-of-Thorns Starfish density.** Marginalised effects of **a** reef zoning and **b** coral cover on density of Pacific Crown-of-Thorns Starfish (CoTS, *Acanthaster* cf. *solaris*) (see "Methods"; Supplementary Method 4). In **a**, $\beta_o$ corresponds to CoTS density on unfished reefs with a hypothetical coral cover = 0; $\beta_1$ constitutes a direct test of whether CoTS densities are higher on fished reefs. In **b**, $\beta_2$ constitutes a direct test of whether CoTS densities decrease with increasing coral cover. Mean posterior predictions presented as points (**a**) and dashed line (**b**), and 95% Bayesian credible intervals (calculated from 20,000 posterior draws) as errors bars (**a**) and grey shaded polygon (**b**). Source data are provided as a Source data file.

are driven at least in part by removal of coral reef fish through commercial and recreational fisheries.

Spatial zoning and marine reserves are used globally to manage fisheries in marine parks[40,46]. Our results demonstrate that fisheries management, in particular the amount of fish biomass removed (Fig. 1) and no-take marine reserves (Fig. 2), has distinct effects on CoTS densities (Fig. 3). Importantly, they indicate that coral reefs with intact predatory fish assemblages can contribute to restricting CoTS densities and reducing CoTS outbreak size and frequency. These results are surprising given that neither fisheries catch limits nor no-take marine reserves are generally developed with any consideration of controlling CoTS outbreaks. For example, the 2004 re-zoning of individual reefs in the GBR Marine Park (Supplementary Table 1) was not affected by their history of CoTS outbreaks[35], nor was the spatial zoning configuration designed to influence CoTS outbreaks[47]. Despite this, our findings clearly demonstrate the applicability of fisheries-based management to prevent CoTS outbreaks in the GBR Marine Park, and the Indo-Pacific more broadly. Furthermore, the efficacy of targeted fisheries-based management may well be much larger if designed with CoTS population dynamics in mind, e.g. through the protection of reefs that are identified as key nodes in CoTS outbreak and spread processes[29,48,49]. Other less-restrictive fisheries management approaches, such as reduced fisheries take of coral reef fish species known to influence CoTS densities (Fig. 1) or temporal closures of reefs to fishing when environmental conditions conducive to outbreaks are predicted, can still contribute substantially to the recovery of reef fish biomass and associated functionality[39,40]. Given that biomass of coral reef fish, including species that consume pelagic and benthic CoTS, within effective no-take marine reserves can recover within 10–15 years (Fig. 2)[39,40], we anticipate that targeted fisheries management will contribute to controlling CoTS outbreaks within two or three decades. Combining this with current CoTS management interventions, such as direct manual control[15] and improving water quality in land-based run-off[21,29], will significantly enhance efforts to support reef restoration and resilience in a warming climate[14]. In summary, targeted fisheries management, including well-designed and enforced no-take marine reserves, offers a tangible and promising contribution to effectively reduce the incidence and impacts of destructive CoTS outbreaks across the Indo-Pacific region.

## Methods

**Large-scale, long-term field data from the GBR Marine Park.** The field data for CoTS, hard coral cover (here referred to as coral cover) and coral reef fish were obtained from the Australian Institute of Marine Science's (AIMS) Long-Term Monitoring Programme (LTMP), while fisheries retained catch data were supplied by the Queensland Department of Agriculture and Fisheries (QDAF). The LTMP has been surveying CoTS populations and coral cover at reefs across the length and breadth of the GBR Marine Park since 1983[50] and has quantified the status and trend of benthic and reef fish assemblages since 1995. Specific examination of the effectiveness of zoning within the GBR Marine Park has also been undertaken[24]. The surveyed reefs are located within zones open to fishing (i.e. General Use, Habitat Protection and Conservation Park) and zones closed to fishing (i.e. Marine National Park Zones, Preservation and Scientific Research Zones) (Supplementary Table 1). The QDAF fisheries data comprise annual retained catch data from the Coral Reef Fin Fish Fishery including commercial, recreational (including charters) and Indigenous fisheries, as well as the Marine Aquarium Fish Fishery (Supplementary Data 1–3). Monthly catch return logbooks became compulsory for all trawlers and line fisheries on 1 January 1988[30]. Retained catch data from each of these fisheries is collected separately and differently by QDAF (please see details below). Use of these data is by courtesy of the State of Queensland, Australia, through the Department of Agriculture and Fisheries.

For both the LTMP and QDAF data, the data sets are chronologically divided into report (LTMP) or financial (QDAF) years, respectively, from 01 July to 30 June. This means that, for instance, the second semester of 2017 belongs to the 2018 report or financial year. Hereafter we will refer to report or financial year as simply year. Below we explain each of these data sets in more detail.

**LTMP CoTS and coral cover data.** LTMP CoTS and coral cover data are available from 1983 to 2020. Both observed CoTS and coral cover data are based on field observations that employ manta tow surveys around the perimeter of each reef following AIMS' Standard Operational Procedure[51]. Within this period, manta tows were conducted once per year but not all reefs were sampled every year. Briefly, manta tow surveys are a broad-scale technique that covers large areas of reef quickly and provides an assessment of broad changes in the distribution and abundance of corals and CoTS. During surveys, two boats each tow an observer clockwise and anti-clockwise around reef perimeters in a series of 2-min tows until they meet at the other end of the reef. Each observer records categorical coral cover (Supplementary Table 8) and the number and size of any CoTS observed (Supplementary Table 9) at the end of each 2-min tow[51]. Manta tow surveys are a non-targeting, rapid assessment method, and therefore it under-samples CoTS individuals that are <15 cm in total body diameter and also under-estimates true population numbers[52]. CoTS density at the individual manta tow level is measured as total number of individuals per 2-min tow. Coral cover at the individual manta tow level is measured as percentage coral cover category per 2-min tow. Categories (in quotes) were converted to proportions, using the mid-points of each category,

for subsequent analyses as follows: "0" = 0, "1" = 0.05, "1L" = 0.025, "1U" = 0.075, "2" = 0.2, "2L" = 0.15, "2U" = 0.25, "3" = 0.4, "3L" = 0.35, "3U" = 0.45, "4" = 0.625, "4L" = 0.5625, "4U" = 0.6875, "5" = 0.875, "5L" = 0.8125, and "5U" = 0.9375. In our statistical analyses, we use both CoTS density and coral cover proportion at the individual manta tow level for a given reef–year combination, as well as averaged across manta tows for a given reef–year combination (please see statistical modelling below).

**LTMP coral reef fish data.** LTMP coral reef fish data are available from 1995 to 2020, with coral reef fish species surveyed following AIMS' Standard Operational Procedure[53]. Here we focus on observations on 56 paired reefs surveyed biennially from 2006 to 2020. These paired reefs are surveyed as part of a dedicated study to examine the effects of the 2004 re-zoning of the GBR Marine Park, with one reef open to fishing (fished) and the other a no-take marine reserve where fishing was prohibited (unfished)[41]. Reef fishes are surveyed in a standard reef slope habitat on the northeast flank of each reef, which is oblique to the prevailing weather conditions and provides consistent relative exposure from which to draw meaningful spatial and zoning comparisons. At each reef, fishes are surveyed along five permanent 50 m transects in each of three sites ($n = 15$ transects reef$^{-1}$ year$^{-1}$). The start and end of each transect is marked with metal stakes, with smaller metal rods spaced every 10 m. Large mobile fishes (Acanthuridae, Chaetodontidae, Labridae (including scarine parrotfishes), Lethrinidae, Lutjanidae, Serranidae, Siganidae and Zanclidae are counted on 5 m wide belts (transect area = 250 m$^2$). Total lengths (TL) of species targeted by fisheries are also recorded. In line with our hypotheses and for our statistical analyses, we retained monitoring information from paired fished and unfished reefs for the following taxa only: Labridae (wrasses), Lethrinidae (emperors), Lutjanidae (tropical snappers), and Serranidae (rockcods) (Supplementary Table 5). The use of standard operating procedures and the high level of training and experience of observers means that there is a high level of accuracy in the data, with little systematic observer bias in counts[54].

**QDAF fisheries data.** Fisheries data for retained catches (in biomass) for the recreational charter and commercial line fisheries within the Coral Reef Fin Fish Fishery, as well as commercial net and trawl fisheries within the GBR Marine Park, were obtained from QDAF in September 2019. The focus was on obtaining retained catch data for individual coral reef fish species that are either known or likely consumers of different life stages of CoTS[24], as well as for large piscivores that may influence CoTS population outbreaks through more complex trophic cascades[35,36]. Retained catch data from the commercial net and trawl fisheries were included based on advice from QDAF as both these fisheries have targeted and reported on some of these coral reef fish species. The commercial net fishery targets several finfish species, using a variety of different net fishing methods[55]. This fishery has reported annual retained catches for tropical snapper from 1989 onwards, and early records also include some retained catches (albeit relatively small) for coral trout and emperor. The commercial trawl fishery currently targets prawns, Moreton Bay bugs and scallop[55], with early logbook records also reporting retained catches for coral trout, emperor, or tropical snapper.

Fisheries data for retained catches were obtained for each of these four fisheries at the smallest spatial (i.e. site, 6-by-6 nautical miles) and temporal (i.e. annual) resolution available, from 1988–1989 (i.e. reporting year 1989) to 2017–2018 (i.e. reporting year 2018). In addition, we also obtained annual total retained catch data for each of these fisheries at these spatial scales, comprising retained catches for all fin fish for charter and commercial line and net fisheries, as well as for prawns, Moreton Bay bugs and scallops for commercial trawl fisheries. For our analyses, the LTMP CoTS and coral cover data sets were aligned both spatially and temporally with this QDAF retained catch data set (please see response of CoTS density to fish biomass removal below).

The extracted QDAF retained catch data set was based on a list of 532 selected fish taxa, with each taxon linked to a unique CAAB code (Codes for Australian Aquatic Biota; https://www.cmar.csiro.au/caab/). QDAF advised that identification of individual species may not be completely reliable in their data set, hence retained catch data were also obtained or estimated for higher taxonomic levels (i.e. families). For example, data were obtained for popular species such as redthroat emperor (*L. miniatus*) and spangled emperor (*L. nebulosus*), as well as for all other emperor species, to estimate a total retained catch for the emperor family (Lethrinidae). We used the CAAB codes to assign each of the fish species caught by the recreational charter and commercial line, net and trawl fisheries to a coral reef fish family. In line with our hypotheses and for our statistical analyses, we focussed on the following six coral reef fish groups, namely, (1) Labridae (wrasses), (2) Lethrinidae (emperors), (3) *L. miniatus* and *L. nebulosus* (redthroat and spangled emperors), (4) Lutjanidae (tropical snappers), (5) Serranidae (rockcods) and (6) *Plectropomus* spp. and *Variola* spp. (coral trout) (Supplementary Table 2). Potential effects of other known predators of pelagic or benthic CoTS, including species of the families Ballistidae (triggerfish), Chaetodontidae (butterflyfish), Diodontidae (Porcupinefish), Haemulidae (grunters), Pomacentridae (damselfish) and Tetraodontidae (Pufferfish) (Table 1)[24], could not be assessed as data on fisheries biomass for these groups were either not available or too limited for our analyses (Haemulidae).

For comparative purposes, annual retained catch data (in numbers) for the recreational and indigenous fisheries for the GBR Marine Park were obtained from QDAF in October 2018, the QFish website (https://qfish.fisheries.qld.gov.au/, accessed February 2019) and associated technical reports[56–59], respectively. In addition, annual retained catch data (in numbers) for the Marine Aquarium Fish Fishery in the GBR Marine Park were provided by QDAF in September 2019 at the smallest spatial (i.e. site, 6-by-6 nautical miles) and temporal (i.e. annual) resolution available, from 1994–1995 (i.e. reporting year 1995) to 2017–2018 (i.e. reporting year 2018). These data were not included in our statistical analyses because (1) the retained catches are recorded in numbers and not in biomass, and (2) they are not available at spatio-temporal resolutions relevant for our study (recreational and Indigenous fisheries only).

**Response of CoTS density to fish biomass removal.** To determine whether CoTS densities can be predicted by the removal of fish biomass, we amalgamated the two quantitatively independent data sets, the AIMS LTMP CoTS and coral cover data set and the QDAF fisheries data set, at the smallest available spatial (i.e. fisheries logbook reporting site, 6-by-6 nautical miles) and temporal resolution (i.e. monitoring or reporting year). First, each LTMP monitoring location was mapped onto Google Earth and subsequently mapped onto the QDAF logbook reporting grids (30-by-30 nautical miles) and sites (6-by-6 nautical miles) for commercial fishing (see https://www.business.qld.gov.au/industries/farms-fishing-forestry/fisheries/monitoring-reporting/requirements/logbook-maps and Supplementary Fig. 1 for example). Each LTMP CoTS monitoring location visited from 1989 to 2019 was then given a unique grid–site code obtained from these logbook maps. This also showed, however, that some LTMP CoTS locations straddled more than one QDAF reporting grid and/or site. Consequently, the extent of each CoTS monitoring location, specifically the North–South and East–West extremes, was checked onto QDAF logbook maps. CoTS monitoring locations were subsequently given the unique grid–site code based on the largest area of reef covered by this code, or, if reef areas were similar across more than one grid and/or site, based on the length of reef perimeter surveyed. One reef was equally divided between two grid sites; QDAF fisheries data were only available for one of these and the reef was assigned to that grid–site code. Finally, locations and associated reef images on Google Earth were double-checked with those provided on the AIMS' Reef Surveys public website (https://apps.aims.gov.au/reef-monitoring/). Any discrepancies were clarified with the LTMP team to ensure that LTMP monitoring locations were assigned to the correct Fisheries' grid–site code.

A total of 2765 LTMP CoTS density records obtained from 1989 to 2019, comprising both CoTS density and coral cover percentage data, were assigned to a unique Fisheries' grid–site code. Further examination of these assignments showed that 564 LTMP records had ≥2 observations in the same grid–site within the same year; these were addressed as follows. First, 142 of these 564 LTMP records comprised observations from different reefs in both fished and unfished zones. In these cases, the LTMP records obtained from reefs in unfished zones were removed ($n = 68$), as these reefs are closed to fishing and would not have been fished. Second, 422 of these 564 LTMP records comprised observations from multiple reefs either in fished or in unfished zones. In these cases, the LTMP records on CoTS density and coral cover percentage were averaged across multiple observations (mostly duplicate, but some triplicate and one quadruplicate observation) across fished or across unfished zones in the same grid–site within the same year. This resulted in a total of 2481 LTMP records assigned to a unique Fisheries' grid–site code, comprising observations on CoTS density and coral cover percentage obtained from 1989 to 2019, and information on zoning status.

Following assignment of the LTMP records to a unique Fisheries' grid–site code, we subsequently aggregated CoTS densities, for each unique site–year combination, to an average across manta tows conducted on fished reefs only and standardized to total number of individual CoTS per minute. Similarly, for each unique site–year combination, retained catches for coral reef fish were aggregated across commercial line, net and trawl and recreational charter fisheries to obtain an estimate of fish biomass removed. We note that these measurements are quantitatively independent because they were collected by different types of surveys (CoTS manta tows vs. fisheries retained catch reports), are within the same 6-by-6 nautical miles logbook reporting site (although not necessarily on the same reef) and are based on different units of measurement (CoTS density is based on time, whereas fish biomass removal is in kilos). Given these caveats, we assume that the data sets are reasonably comparable for the purposes of the large-scale and long-term trends that are of interest here; however, one should exercise caution when interpreting the results. The merged data set contained 19,239 paired CoTS–fisheries observations collected from 157 individual fisheries logbook reporting sites across the GBR from 1990 to 2018.

We analysed this data set using a Bayesian hurdle-gamma modelling approach, with the probability of CoTS density at time $t + x$ being zero following a Bernoulli distribution with a logit link and the positive continuous outcomes being modelled using a Gamma distribution with a log link. We added coral cover (continuous: proportion) as a fixed effect for both the Bernoulli (zero) and Gamma (non-zero) components of the model in order to maintain the consistency with the two tests conducted under the role of zoning and coral cover on CoTS density. Fisheries in the GBR are only permitted in open reefs, so paired CoTS–fisheries observations where CoTS data came from closed zones only were removed and zoning status was not included as a covariate in this model. Because fish effects on CoTS density might be manifested after multiple time lags[28], we ran six models for each of the six coral reef fish

groups of interest, each corresponding to a $x$ time lag between fish biomass removal and CoTS density from 1 through 6 years (i.e. 36 models in total). For the Gamma component only, and in order to examine whether CoTS densities within sites increase with increasing removal of coral reef fish following multiple time lags, we also added annual fish biomass removal (continuous: kg) at time $t$ as a fixed effect; grid–site ID was added as having a hierarchical effect on the model intercept. We note that coral cover data were from the same years as CoTS density data (i.e. $t + x$). In Supplementary Method 2, we provide detailed model and algorithm fitting specifications, a power analysis to check predictive power of original model, posterior predictive checks, comparisons between prior and posteriors distributions, posterior distribution of model parameters and chain mixing trace-plots (Supplementary Figs. 4–40). We then determined the percentage of the posterior distribution of the slope between annual fish biomass removal at time $t$ and CoTS density at time $t + x$ that falls above 0.

**Effects of no-take marine reserves on coral reef fish**. To compare the biomass, density and size of the six fish groups on fished and unfished reefs, specifically those groups that influence CoTS densities, we used LTMP coral reef fish observations from 840 transects conducted on 56 paired fished and unfished reefs along the length of the GBR Marine Park between 2006 and 2020. All LTMP reef fish data from all species monitored within the families Labridae (wrasses—five species), Lethrinidae (emperors —16 species), Lutjanidae (tropical snappers—21 species) and Serranidae (rockcods— 35 species) (Supplementary Table 5) were standardized by converting raw counts to densities per 1000 m$^2$. Size was analysed at the site level as mean population total body length (TL cm). Standing biomass (kg) per 1000 m$^2$ was calculated for each fish species from estimated fish lengths using published length–weight relationships[60,61]. Inferences about differences between no-take marine reserves and reefs open to fishing were based on Bayesian C.I.s calculated from parameters estimated from hierarchical models. Site-level mean population length was modelled following a gamma distribution, whereas density and standing biomass were modelled with a hurdle-gamma approach, similarly to described above for fisheries-dependent data. Each response was modelled as a function of zoning status (fished vs. unfished, categorical fixed effect), while accounting for the effects of site, reef pair ID, reef ID and year, following Emslie et al.[41]. In the hurdle-gamma models, the logistic component was modelled as a function of zoning status. In Supplementary Method 3, we provide detailed model and algorithm fitting specifications, posterior predictive checks, comparisons between prior and posteriors distributions, posterior distribution of model parameters and chain mixing trace-plots (Supplementary Figs. 41–60). Differences between values for unfished and fished reefs were then expressed as a percentage of the value on the fished reefs, such that a higher value in unfished compared with fished reefs would yield a positive difference, whereas a lower value would give a negative difference.

**Effects of no-take marine reserves and coral cover on CoTS density**. To quantify the effects of reef zoning and coral cover on CoTS density, we used CoTS density and proportional coral cover observations from 3358 manta tow surveys (i.e. reef–year combinations) conducted on 490 reefs along the length of the GBR Marine Park between 1985 and 2020. This data set contains 157,348 individual 2-min manta tows in which a total of 52,921 individual CoTS were recorded. Specifically, we employed a Bayesian hierarchical approach to model CoTS density per tow (individuals/2 min) following a negative binomial distribution. First, to determine whether CoTS densities are higher on fished reefs compared to unfished reefs we added zoning status (categorical: open, closed) as a fixed effect. Second, to determine whether CoTS densities change with coral cover we added tow-level coral cover (continuous: proportion) as a fixed effect. Reef–year combination ($n = 3358$) was added as an intercept-level hierarchical effect. In Supplementary Method 4, we provide detailed model and algorithm fitting specifications, posterior predictive checks, comparisons between prior and posteriors distributions, posterior distribution of model parameters and chain mixing trace-plots (Supplementary Figs. 61 and 62). We also tested for but found no evidence of zero inflation in the data (Supplementary Method 4 and Supplementary Fig. 63).

**Testing hypotheses**. For most models, we assess each hypothesis against the posterior distribution of its corresponding parameter in terms of percentage probabilities. For example, in the hurdle-gamma models described in the section above, we measure the percentage of the posterior distribution of the slope between annual fish biomass removal at time $t$ and CoTS density at time $t + x$ that falls above 0.

**Reporting summary**. Further information on experimental design is available in the Nature Research Reporting Summary linked to this paper.

## Data availability
The data that support the findings of this study are available from the Australian Institute of Marine Science (AIMS) and Queensland Department of Agriculture and Fisheries (QDAF). The AIMS Long Term Monitoring Programme data are available online: https://eatlas.org.au/gbr/ltmp-data. The QDAF data are owned by the Department and AIMS accessed and used the QDAF data under a data transfer agreement. Request for the QDAF data can be made to the authors with a 1-month time frame for a response to

requests. The CAAB code represents Codes for Australian Aquatic Biota (CAAB) number and is available online: https://www.cmar.csiro.au/data/caab/. Source data are provided with this paper.

## Code availability
The R Code and part of the data used in our study can be downloaded from https://github.com/open-AIMS/cots_fish_trends. If using our code and/or data, please cite it as: Kroon FJ, Barneche DR, Emslie MJ (2021) open-AIMS/cots_fish_trends: Accepted version of paper data and code of manuscript: Fish predators control outbreaks of Crown-of-Thorns Starfish (Nature Communications)[62]. Zenodo: https://doi.org/10.5281/zenodo.5560820.

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

## Acknowledgements

We are indebted to the Queensland Department of Agriculture and Fisheries for sharing fisheries data for the Coral Reef Fin Fish Fishery and the Marine Aquarium Fish Fishery in the GBR Marine Park. Use of the data is by courtesy of the State of Queensland, Australia through the Department of Agriculture and Fisheries. We thank David Westcott for constructive discussions during the project, Murray Logan for statistical advice and Aaron MacNeil for comments on the manuscript. This study and its publication were supported by the Australian Institute of Marine Science. The authors acknowledge the Wulgurukaba and Bindal people, Traditional Owners of Townsville and the Whadjuk Noongar people, Traditional Owners of Perth, where this study was conducted.

## Author contributions

F.J.K. conceptualized and coordinated the study, obtained fisheries data from Queensland Department of Agriculture and Fisheries (QDAF), aligned AIMS Long-Term Monitoring Programme (LTMP) CoTS and coral data with QDAF fisheries data and led the writing and preparation of the manuscript. M.J.E. extracted CoTS, coral cover and coral reef fish data from the AIMS LTMP database. D.R.B. organized AIMS LTMP CoTS, coral cover and coral reef fish monitoring data and QDAF fisheries data for each of the analyses, conducted the statistical analyses and prepared graphics. All authors contributed to data and results interpretation and to writing subsequent manuscript drafts.

## Competing interests

The authors declare no competing interests.
