## [Peer Review File · Nature Communications]

Reviewer comments, first review round:

Reviewer #1 (Remarks to the Author):

Dear authors

Thank you for the opportunity to review your paper. I really enjoyed reading it, well written, interesting, and highly topical subject.

I have suggested a few edits, but nothing major

I would offer that you could make this a stronger paper by exploring why we manage COTS the way we do, and why this type of work demonstrates that COTS management, or our coral reef management requires adaptive and integrative management of multiple pressures and impacts. Our understanding of COTS and predation has been very driven by a limited understanding of what can prey on COTS, and your paper explores how important to have a fully functional coral reef system, with prey-predation in place. My only criticism, if you can call it that, is that you do not explore this functionality enough in the text. It would not take much to include a few more lines around this. The ability for a reef to function (and our ability to understand it) needs to include studies that work across many trophic levels. This paper highlights that fish eat COTS, and that in protected areas, more fish = less COTS. This seems so logical but yet, has never been part of the COTS management priorities because we tend to manage COTS based on distribution and abundance, not as part of a fully adaptive management plan.

In Line 41 - you discuss COTS being a contentious issue between source (nutrients, food supply, increased survival of COTS) or overfishing. You present this as an either/or issue and it is far more than that. I think this type of work points to the concept of management of multiple pressures for multiple benefits. Impact on larval recruitment also impacts on fish communities, so lots of interactions here.

You would still need to consider management of all these issues. There is no one solution, and reduction of fishing pressures plus reduction of land based pollution would achieve multiple benefits, and I would like to see this considered here.

Line 56 - you discuss the type of fish that prey on COTS, and note that predation is higher than "currently appreciated". It would be good to state what that is - what is our current understanding of how many COTS are eaten/preyed upon by fish.

There is also discussion that predation will most likely happen at the larval or benthic state (juvenile). You do note that it's very difficult to have detailed information due to lack of fisheries data, but can you expand on this at all?. Are certain types of fish only eating the larval state. If you have more of those fish, is this a better constraint to COTS outbreaks than partial or full predation at the benthic stage. Is this something that could be explored in future work - as would be useful to consider when identifying possible areas/activities for MPAs?

All the fish in Extended Table 1 prey on COTS, but are some more opportunistic than others, and others in which COTS are an important food source. If so, could COTS then drive recovery of some fisheries if providing a food source in areas of depleted reefs? I recognise this is a difficult question to answer so perhaps just an extended sentence around what isn't known in terms of fisheries (Line 61).

Line 99 - this whole line is difficult to follow, not sure of the term "release" in this context.

I have no issues with the statistical analysis of data, it is a strong approach, using long term data, and supports the basis of your study, that overfishing can increase COTS numbers. Can you test or have tested that reefs with high number of COTS may already have lost some functionality, including food source for some of the fish species you identify, and that has also impacted on the fish numbers?

Line 112 - just back to my original point, fisheries based management may reduce COTS, but it's not the only tool in the tool box, and should be considered as part of an integrative approach of MPAs, pollution reduction, fisheries protection.

Excellent paper, I enjoyed reading it very much, and hope that this goes a long way in demonstrating connectivity in all of our decisions within coral reef management

Reviewer #2 (Remarks to the Author):

Crown-of-Thorns starfish (CoTS) outbreaks are a major contributor to coral loss across the Indo-Pacific Oceans and the underlying causes for population outbreaks have been subject to debate since the first recorded outbreak in the 1960s. Given the increasing threats posed to coral reef systems, it is critically important to assess whether human interference is contributing to CoTS outbreaks.

Kroon et al. have used extensive long-term data sets covering the length of the GBR, which provide a unique opportunity to address the ongoing debate on whether predator removal contributes to CoTS outbreaks. I commend the authors in their efforts to shed more light on this important subject. I do not fully understand the statistical methods used in this manuscript, which in themselves appear appropriate and sound, but I do have certain questions and concerns that need to be clarified or addressed with respect to the data used and their interpretation.

My main concerns relate to the first part of the study. The authors have undertaken the ambitious task of combining Queensland Department of Agriculture and Fisheries (QDAF) data with long-term monitoring program (LTMP) coral cover and CoTS density data. As the authors acknowledge in their manuscript, there were a number of challenges in merging QDAF fisheries data with LTMP coral cover and CoTS density data. The grid used in the QDAF data set is relatively coarse and often includes multiple reefs, which although taken into account in the analyses, may affect the quality of the data. More specifically, please address or comment on the following points:

- Biomass removal of potential CoTS predators is an interesting approach, but it provides no indication or quantitative measure on the standing abundance or biomass of the fish groups used in the study. Why not use the fish data from the LTMP data set instead?
- The authors report strong effects of predator removal on CoTS density after a 1, 2 and 4 year time lag. However, I am concerned about the mostly opposite effect observed for the 3 year time lag. This is not mentioned in the text. How do the authors explain these inconsistencies? (Line 103)
- It is my understanding that CoTS densities were analysed separately for the 6 fish groups. Have the authors considered interactions between the fish groups? It would be critically important to know whether for example levels of coral trout biomass removal coincide with similar levels of removal in lethrinids or other serranids. If this is the case, the increased CoTS densities may not be caused by the trophic cascade effects of coral trout, but rather the removal of known predators such as redthroat or spangled emperor.

The second part of the study investigates the effects of no-take marine reserves on the biomass, density and length of the same 6 fish groups used in the first part. Again, I wonder why this data was not used to directly determine the effects of predator abundance on CoTS densities.

The third part of the study shows a clear increase in CoTS densities on reefs open to fishing compared to those closed to fishing. Given CoTS densities decreased with coral cover, it would be interesting to see the same comparison between open and closed reefs for coral cover to ascertain it is a fisheries-based mechanism.

Minor comments:

Line 78- specify that this refers to the commercial catch?

Line 101 - Please elaborate on what you mean by behavioural release of planktivorous fishes allowing CoTS larvae to reach the benthos.

Line 450 - I am curious to know why the authors chose to include net and trawl fishery data, assuming there would be little interaction between CoTS / reefs and taxa caught. How do the

results compare when this data is excluded from the analyses?

Line 494 - Correct LMTP

Line 515 - "...that that..."

Line 549 - Presumably these are the "paired" reefs mentioned in the results? To provide more clarity in the methods, I suggest expanding the description of the reefs used here.

Supp Table S4 - Are the 5 listed labrid species the only ones monitored in the LTMP? If not, why were only these five species chosen for the analyses, as opposed to all those listed as targeted by the fishery? Presumably, the main labrids caught since the ban on *Cheilinus undulatus* would be the larger tusk fish species?

Reviewer #3 (Remarks to the Author):

The paper makes a statistical case for the "predator removal" hypothesis that "release from predation pressure is a primary cause of CoTS outbreaks," and also finds evidence of "involvement of more complex ecological effects in modulating CoTS outbreaks". It's an important finding, and the authors' suggestion that CoTS population dynamics be factored into fisheries management could be influential.

The statistical analyses involve Bayesian hierarchical modelling. It is competently done: the models follow well established procedures, the choices for distributions (including priors) are natural, and the conclusions are appropriate. Some additional analyses would be useful.

1) The authors should investigate the robustness of the conclusions to choice of prior, as this is easy to do, and it is important to understand the influence of the prior.

2) It would be useful to apply the methods to simulated data to demonstrate that the methods can recover true parameters. Simulated data would also be useful to assess the impact of recreational fishing. The authors are surely correct that: "These effects are likely stronger, given that the take of emperors, tropical snappers and rockcods by the recreational (non-charter) fisheries in the GBR Marine Park is estimated to have been of similar magnitude to that of the commercial line fisheries for decades". However, it would be useful to investigate this claim using simulated data.

3) The authors might want to consider applying methods of function regression to this data set, instead of performing separate analyses for each lag x . Function regression allows the dependent and/or independent variable to be a function. Here the CoTS density as a function of lag x , and/or the biomass extraction as a function of t , could be modelled in this manner. This could improve precision if either function varies smoothly. I do not expect the authors to take up this suggestion for this paper, but they might.

A few other minor comments:

1) Line 51: "a 100" should just be "100"

2) Extended Table 1 caption should have superscript '#' on "Parrotfish".

3) Line 33 of supplementary information uses label S1, all other uses exclude the 'S'.

4) Line 119 of supplement: density measured in individuals per minute instead of individuals per 2 minute tow.

5) Line 144: the authors use the term 'warm-up' to describe what is more commonly referred to as 'burn-in'.

6) Why was zoning status not included as a covariate on Line 533? A brief explanation would suffice.

REVIEWER COMMENTS

Reviewer #1 (Remarks to the Author):

Dear authors

Thank you for the opportunity to review your paper. I really enjoyed reading it, well written, interesting, and highly topical subject. I have suggested a few edits, but nothing major.

Response: Thank you for your positive comments and constructive feedback.

- 1. I would offer that you could make this a stronger paper by exploring why we manage COTS the way we do, and why this type of work demonstrates that COTS management, or our coral reef management requires adaptive and integrative management of multiple pressures and impacts. Our understanding of COTS and predation has been very driven by a limited understanding of what can prey on COTS, and your paper explores how important to have a fully functional coral reef system, with prey-predation in place. My only criticism, if you can call it that, is that you do not explore this functionality enough in the text. It would not take much to include a few more lines around this. The ability for a reef to function (and our ability to understand it) needs to include studies that work across many trophic levels. This paper highlights that fish eat COTS, and that in protected areas, more fish = less COTS. This seems so logical but yet, has never been part of the COTS management priorities because we tend to manage COTS based on distribution and abundance, not as part of a fully adaptive management plan.*

Response: The reviewer makes an important point about the importance of managing CoTS as part of a fully functional coral reef system. We have addressed this as follows:

1. We have included a paragraph in the Introduction on current management approaches to CoTS population outbreaks (pl see our detailed response to this Reviewer's 2nd comment below).

2. We have added a few lines in the Discussion to recommend future research on this functionality, as follows (new text in bold):

*'Critically, our findings support the importance of hypothesised ecological and behavioural release^{35, 36}, following removal of the piscivorous coral trout, in influencing CoTS population dynamics and outbreaks. **The involvement of more complex ecological effects in modulating CoTS outbreaks²¹ have remained speculative until now; our results warrant further research to better understand and quantify the potential role of such trophic cascades.'***

- 2. In Line 41 - you discuss COTS being a contentious issue between source (nutrients, food supply, increased survival of COTS) or overfishing. You present this as an either/or issue and it is far more than that. I think this type of work points to the concept of management of multiple pressures for multiple benefits. Impact on larval recruitment also impacts on fish communities, so lots of interactions here. You would still need to consider management of all these issues. There is no one solution, and reduction of fishing pressures plus reduction of land based pollution would achieve multiple benefits, and I would like to see this considered here.*

Response: The reviewer is correct in that the text in the original manuscript may have read as an either/or issue between the two hypotheses, even though this was not our intention. Rather,

our intention was to express exasperation about the fact that arguing about one or the other has been going on for over 50 years, seriously hampering any progress on managing the detrimental impacts of CoTS outbreaks on coral reefs. Re-reading we appreciate that this is not how this may have come across. We have addressed this by adding (i) a clarifying sentence in the first paragraph of the Introduction, and (ii) a new 2nd paragraph presenting contemporary management approaches for CoTS, as follows (changes in bold):

‘[...] Whether human interference has exacerbated CoTS outbreaks, through increasing nutrient levels enhancing larval recruitment¹¹ or harvesting of natural predators that would limit CoTS abundance⁹, remains a contentious topic². **While neither hypothesis has received universal or unequivocal support², their contributions to causing or propagating CoTS outbreaks are also not mutually exclusive.** Given that CoTS outbreaks continue to be one of the major drivers of coral loss¹², including during recent mass bleaching events¹³, new pathways for CoTS control at large scale have become increasingly important to halt further declines in coral cover and support reef restoration and resilience in a warming climate across the Indo-Pacific¹⁴.

Contemporary management to reduce the detrimental impact of CoTS population outbreaks on coral reefs centres around a combination of direct manual control and water quality improvement^{12, 15}. Manual control programs have been implemented since the 1960s^{7, 16}, killing or removing an estimated 17 million CoTS across the Indo-Pacific by 2014², but with limited success in reducing either CoTS densities or coral loss¹⁷. Recent improvements in single-injection methods to cull individual CoTS¹⁸, combined with strategic deployment of manual control¹⁹, have significantly improved the efficacy of manual control programs to reduce CoTS impacts on coral reefs^{14, 15}. The role of water quality improvement programs in CoTS control are predicated on the hypothesized link between nutrient enrichment and CoTS outbreaks¹¹. The ‘*nutrient enrichment*’ hypothesis states that high nutrient availability increases phytoplankton biomass and enhances CoTS larval growth and survival leading to mass recruitment events and outbreaks. In Australia, the purported role of nutrient enrichment from land-based run-off (i.e. the ‘*terrestrial run-off*’ hypothesis)¹¹ has become a central argument for policy and investment to improve water quality of the Great Barrier Reef (GBR) World Heritage Area²⁰. While its efficacy is supported by independent modelling exercises^{14, 21}, catchment and water quality improvement efforts, implemented since the early 2000s²², are unlikely to have acted to suppress CoTS population dynamics and cannot yet be relied upon to contribute to CoTS control on the GBR¹⁵.

Please note that in the Discussion in the original manuscript, we already concluded, based on our findings, that fisheries management can *contribute* to controlling CoTS outbreaks, *in combination with* current CoTS management interventions:

‘[...] Given that biomass of coral reef fish, including CoTS predators, within effective no-take marine reserves can recover within 10 to 15 years (Fig. 2)^{38, 39}, we anticipate that targeted fisheries management will contribute to controlling CoTS outbreaks within two or three decades. Combining this with current CoTS management interventions such as direct manual control¹⁵ and improving water quality in land-based run-off^{21, 29}, will significantly enhance efforts to support reef restoration and resilience in a warming climate¹⁴. In summary, fisheries management, including well-designed and enforced no-take marine reserves, offers a tangible and promising contribution to effectively reduce the incidence and impacts of destructive CoTS outbreaks across the Indo-Pacific region.’

3. *Line 56 - you discuss the type of fish that prey on COTS, and note that predation is higher than "currently appreciated". It would be good to state what that is - what is our current understanding of how many COTS are eaten/preyed upon by fish.*

Response: We have adjusted this sentence to clarify our statement (changes in bold):

Original: Recent detection of CoTS DNA in faecal and gut content samples from wild-caught coral reef fish further indicate that direct fish predation on pelagic and benthic CoTS might be more common than is currently appreciated⁸.

Revised: **Recently, CoTS DNA was detected in faecal and gut content samples from 18 wild-caught coral reef fish species, including nine fish species which had not previously been reported to feed on CoTS¹⁶. This further indicates that more coral reef fish species might predate on pelagic and benthic CoTS than is currently appreciated.**

4. *There is also discussion that predation will most likely happen at the larval or benthic state (juvenile). You do note that it's very difficult to have detailed information due to lack of fisheries data, but can you expand on this at all? Are certain types of fish only eating the larval state. If you have more of those fish, is this a better constraint to COTS outbreaks than partial or full predation at the benthic stage. Is this something that could be explored in future work - as would be useful to consider when identifying possible areas/activities for MPAs? All the fish in Extended Table 1 prey on COTS, but are some more opportunistic than others, and others in which COTS are an important food source. If so, could COTS then drive recovery of some fisheries if providing a food source in areas of depleted reefs? I recognise this is a difficult question to answer so perhaps just an extended sentence around what isn't known in terms of fisheries (Line 61).*

Response: First, we have rephrased the last sentence of the 2nd last paragraph in the Introduction to better place this sentence in the context of the 'predator removal' hypothesis (i.e. the topic of this paragraph; changes in bold):

'While independent modelling studies have provided support for coral reef fish predation regulating CoTS outbreaks^{26, 27, 28, 29}, the lack of **data on harvesting of natural predators**, particularly at spatio-temporal scales large enough to encompass CoTS outbreak dynamics, means we have limited understanding of how **removal of coral reef fish may** affect CoTS abundance and the applicability of **predator-based management** to prevent CoTS outbreaks.'

Second, the reviewer is quite right that CoTS predation can occur at the larval and benthic life stage. Reported predation by coral reef fish families on larval and benthic CoTS was originally summarised in Extended Data Table 1; we have now brought this Table into the main manuscript as Table 1. In our study, we were only able to examine the effect of fish biomass removal of those species that have been reported to consume benthic CoTS. In the Results ('Response of CoTS density to fish biomass removal'), we have now included the following two sentences (in bold) to clarify which fish groups we focussed on, and which ones we could not assess due to unavailability of fisheries biomass data, including families reported to consume pelagic CoTS (butterfly fish and damselfish):

'Coral reef fisheries may affect CoTS density through the removal of fish species that either directly predate on²⁴, or indirectly influence predation on CoTS²¹. Here, we focussed on six fish groups based on their reported consumption of benthic CoTS (Table 1)²⁴ and their contribution to the commercial and recreational charter fisheries in the GBR Marine Park (Supplementary Tables 2-3)³². **These six groups comprised (1) Labridae (wrasses), (2)**

Lethrinidae (emperors), (3) *Lethrinus miniatus* and *L. nebulosus* (redthroat and spangled emperors), (4) Lutjanidae (tropical snappers), (5) Serranidae (rockcods) and (6) *Plectropomus* spp. and *Variola* spp. (coral trout). Potential effects of other known predators of pelagic or benthic CoTS, including species of the families Ballistidae (triggerfish), Chaetodontidae (butterflyfish), Diodontidae (Porcupinefish), Haemulidae (grunters), Pomacentridae (damsel fish) and Tetraodontidae (Pufferfish) (Table 1)²⁴, could not be assessed as data on fisheries biomass for these groups were either not available, or too limited for our analyses (Haemulidae).'

Third, we expanded briefly in one of our concluding sentences to clarify that not only does fish biomass of CoTS predators increase in effective no-take marine reserves, but this has been reported for fish species that consume pelagic and benthic CoTS (changes in bold):

‘Given that biomass of coral reef fish, including **species that consume pelagic and benthic** CoTS, within effective no-take marine reserves can recover within 10 to 15 years (Fig. 2)³⁸,³⁹, we anticipate that targeted fisheries management will contribute to controlling CoTS outbreaks within two or three decades.’

Finally, whether CoTS is a more important food source for some fish species compared to others is completely unknown. Based on current evidence, none of the fish species known to consume CoTS (pelagic or benthic) are completely reliant on CoTS, i.e. all appear to be opportunistic predators. Earlier modelling studies suggested that none of the (then) known fish predators would be completely reliant on CoTS, as this would not be sustainable during non-outbreak conditions when CoTS outbreaks are very low. Notwithstanding, it is exactly during those conditions that fish predation is thought to be critical to prevent CoTS outbreaks from occurring (as discussed in the 2nd paragraph of the Discussion). Whether CoTS may drive recovery of some fisheries by providing a food source is an interesting concept, but somewhat outside the scope of our study.

5. *Line 99 - this whole line is difficult to follow, not sure of the term "release" in this context.*

I have no issues with the statistical analysis of data, it is a strong approach, using long term data, and supports the basis of your study, that overfishing can increase COTS numbers. Can you test or have tested that reefs with high number of COTS may already have lost some functionality, including food source for some of the fish species you identify, and that has also impacted on the fish numbers?

Response: We have changed this line and elaborated to clarify what we mean by the term “release”; please see our response to Reviewer 2, comment 7.

We appreciate the reviewer’s suggestion to test whether reefs with high number of CoTS may already have lost some functionality. We would argue that by showing an increase in CoTS densities with increasing biomass removal of coral reef fish, we have demonstrated a critical loss of functionality that is of particular interest for our study. That is, increasing biomass removal of coral reef fish results in reduced consumption by coral reef fish *known to predate on CoTS*. The information on coral reef fish known to consume CoTS (and targeted by fisheries in the GBR) was previously presented in Extended Data Table 1; we have moved this Table into the main manuscript (Table 1) to present upfront that many coral reef fish families have been reported to consume CoTS, including ones that are targeted by fisheries. Removal of these coral reef fish families by fisheries (at least for those for which data was available) directly resulted in an increase in CoTS densities. We have also included the

following text in the Results ('Response of CoTS density to fish biomass removal') to make absolutely clear which fish groups we have focussed on and why:

Here, we focussed on six fish groups based on their reported consumption of benthic CoTS (Table 1)²⁶ and their contribution to the commercial and recreational charter fisheries in the GBR Marine Park (Supplementary Tables 2 and 3)³⁵. These six groups comprised (1) Labridae (wrasses), (2) Lethrinidae (emperors), (3) *Lethrinus miniatus* and *L. nebulosus* (redthroat and spangled emperors), (4) Lutjanidae (tropical snappers), (5) Serranidae (rockcods) and (6) *Plectropomus* spp. and *Variola* spp. (coral trout). Potential effects of other known predators of pelagic or benthic CoTS, including species of the families Ballistidae (triggerfish), Chaetodontidae (butterflyfish), Diodontidae (Porcupinefish), Haemulidae (grunters), Pomacentridae (damselfish) and Tetraodontidae (Pufferfish) (Table 1)²⁶, could not be assessed as data on fisheries biomass for these groups were either not available, or too limited for our analyses (Haemulidae).

Our findings also provide support for the involvement of more complex ecological effects in modulating CoTS outbreaks, which have remained speculative until now. In the Discussion, we have included a recommendation for further research on these more complex ecological effects, as follows (changes in bold):

Original: Critically, our findings support the importance of hypothesised ecological and behavioural release^{37, 38}, following removal of the piscivorous coral trout, in influencing CoTS population dynamics and outbreaks.

Revised: Critically, our findings support the importance of hypothesised ecological and behavioural release^{37, 38}, following removal of the piscivorous coral trout, in influencing CoTS population dynamics and outbreaks. **The involvement of more complex ecological effects in modulating CoTS outbreaks²³ have remained speculative until now; our results warrant further research to better understand and quantify the potential role of such trophic cascades.**

6. *Line 112 - just back to my original point, fisheries based management may reduce COTS, but it's not the only tool in the toolbox, and should be considered as part of an integrative approach of MPAs, pollution reduction, fisheries protection.*

Response: Please see our response to this Reviewer's first and second comments.

7. *Excellent paper, I enjoyed reading it very much, and hope that this goes a long way in demonstrating connectivity in all of our decisions within coral reef management*

Response: Thank you! We're glad you enjoyed reading it.

Reviewer #2 (Remarks to the Author):

Crown-of-Thorns starfish (CoTS) outbreaks are a major contributor to coral loss across the Indo-Pacific Oceans and the underlying causes for population outbreaks have been subject to debate since the first recorded outbreak in the 1960s. Given the increasing threats posed to coral reef systems, it is critically important to assess whether human interference is contributing to CoTS outbreaks.

Kroon et al. have used extensive long-term data sets covering the length of the GBR, which provide a unique opportunity to address the ongoing debate on whether predator removal contributes to CoTS outbreaks. I commend the authors in their efforts to shed more light on this important subject. I do not fully understand the statistical methods used in this manuscript, which in themselves appear appropriate and sound, but I do have certain questions and concerns that need to be clarified or addressed with respect to the data used and their interpretation.

Response: Thank you for your positive comments and constructive feedback. We have addressed your questions and concerns in more detail below.

- 1. My main concerns relate to the first part of the study. The authors have undertaken the ambitious task of combining Queensland Department of Agriculture and Fisheries (QDAF) data with long-term monitoring program (LTMP) coral cover and CoTS density data. As the authors acknowledge in their manuscript, there were a number of challenges in merging QDAF fisheries data with LTMP coral cover and CoTS density data. The grid used in the QDAF data set is relatively coarse and often includes multiple reefs, which although taken into account in the analyses, may affect the quality of the data. More specifically, please address or comment on the following points:*

Biomass removal of potential CoTS predators is an interesting approach, but it provides no indication or quantitative measure on the standing abundance or biomass of the fish groups used in the study. Why not use the fish data from the LTMP data set instead?

Response: We understand the point the referee is making, however, we do not necessarily agree for the following two reasons.

First, our study is specifically focussed on whether *removal of predators* (i.e. coral reef fish) influenced recent population outbreaks of CoTS in the GBR Marine Park. Given that commercial and recreational fisheries have been the major extractive activities on the GBR since at least the 1950s, we were specifically interested in the removal of coral reef fish through these fisheries and used QDAF fisheries data rather than the standing abundance or biomass of the fish groups.

Second, the spatial scales of the QDAF fisheries data and the LTMP CoTS data are more similar than the LTMP fish data and LTMP CoTS data. The QDAF fisheries data were obtained at the smallest spatial scale available (i.e. site, six by six nautical miles) and are for reefs open to fishing within these 6 x 6 nautical mile logbook reporting sites only. The LTMP CoTS data are based on observations around the perimeter of each individual reef. In contrast, the LTMP fish data are based on observations at a much smaller spatial scale, i.e. five permanent 50 m transects in each of three sites ($n = 15$ transects reef⁻¹ year⁻¹), with large mobile fishes counted on 5 m wide belts (transect area = 250 m²).

Combined, we therefore argue that for the purpose of the large-scale and long-term trends of interest to our study, the QDAF fisheries data and the LTMP CoTS data are more suitable and comparable than the LTMP fish data and LTMP CoTS data.

2. *The authors report strong effects of predator removal on CoTS density after a 1, 2 and 4 year time lag. However, I am concerned about the mostly opposite effect observed for the 3 year time lag. This is not mentioned in the text. How do the authors explain these inconsistencies? (Line 103)*

Response: We thank the referee for their comment but would like to disagree that the 3-year time lag shows a mostly opposite effect. We find strong positive effects of biomass removal on CoTS densities (both >80% and >95% probabilities) for five fish groups following time lags of 1, 2, and 4 years. Such strong effects are less common following time lags of 3, 5 and 6 years, but the effects are not the opposite for year 3 (i.e. they are not either <20% or <5% probabilities) (pl see also Supplementary Figure 2). In the Results, we have added the following text to this line (new text in bold):

“Across these five fish groups, the **positive** effects of biomass removal on CoTS densities were most pronounced (**both >80% and >95% probabilities**) following time lags of 1, 2 and 4 years, **but less common following time lags of 3, 5 and 6 years** (Fig. 1; Supplementary Fig. 2; Supplementary Table 4).”

3. *It is my understanding that CoTS densities were analysed separately for the 6 fish groups. Have the authors considered interactions between the fish groups? It would be critically important to know whether for example levels of coral trout biomass removal coincide with similar levels of removal in lethrinids or other serranids. If this is the case, the increased CoTS densities may not be caused by the trophic cascade effects of coral trout, but rather the removal of known predators such as redthroat or spangled emperor.*

Response: We thank the referee for their comment. We now provide information on potential interactions between the six fish groups through pairwise Pearson correlation values for fish biomass removal for each of the six time lags (Supplementary Table 6). In the Result section, we elaborate on the meaning of these correlation values and the potential implications for trophic cascade effects, as follows (new text in bold):

“**Our finding that CoTS densities increased with increasing biomass removal of coral trout is unexpected given that adult coral trout are almost entirely piscivorous³³. This result may be related to levels of coral trout biomass removed corresponding with similar levels of biomass removed in other fish groups. Fish biomass removed did indeed coincide strongly between coral trout and Serranidae (pairwise Pearson correlation values >0.900 for all six time lags), and between redthroat and spangled emperors (*Lethrinus miniatus* and *L. nebulosus*) and Lethrinidae (>0.900 for four of six time lags) (Supplementary Table 6). This aligns with the commercial line fishery primarily targeting coral trout and redthroat emperor, with another 20 species targeted including emperor, tropical snapper and rockcod species (Supplementary Table 3)³². In contrast, fish biomass removed coincided less strongly between coral trout and redthroat and spangled emperors (range of pairwise Pearson correlation values: 0.155 – 0.650), and between coral trout and Lethrinidae (0.564 – 0.678) (Supplementary Table 6). Thus, while some interaction between the different fish groups cannot be discounted, the positive effects of coral trout biomass removed by fisheries on CoTS densities can**

also not be completely explained by these interactions. While CoTS consumption has not been reported for coral trout, juvenile *P. leopardus* feed mainly on benthic invertebrates³⁴ and further research on whether this may include recently settled or juvenile CoTS is warranted. If juvenile coral trout indeed consume CoTS, removal of adult coral trout may indirectly influence predation on CoTS given the importance of high levels of self-recruitment in maintaining *Plectropomus* populations³³. Finally, our finding could support the involvement of more complex ecological effects in modulating CoTS outbreaks²¹, which have remained speculative until now. Such effects may include the ecological release of invertebrates that prey on juvenile CoTS, when higher numbers of large piscivores result in reduced densities of benthic carnivorous fishes³⁵. It could also include the behavioural release of planktivorous fishes away from the reef substrate with reduced numbers of large piscivores, resulting in increased number of CoTS larvae able to reach the benthos³⁶.”

4. *The second part of the study investigates the effects of no-take marine reserves on the biomass, density and length of the same 6 fish groups used in the first part. Again, I wonder why this data was not used to directly determine the effects of predator abundance on CoTS densities.*

Response: Please see our response to this Reviewer’s comment #1 above.

5. *The third part of the study shows a clear increase in CoTS densities on reefs open to fishing compared to those closed to fishing. Given CoTS densities decreased with coral cover, it would be interesting to see the same comparison between open and closed reefs for coral cover to ascertain it is a fisheries-based mechanism.*

Response: We understand the point the referee is making. However, we have already demonstrated that CoTS density declines with increasing coral cover after accounting for the effects of zoning status (Fig. 3b). Moreover, given that CoTS density increases on open reefs (Fig. 3a, corrected for a fixed coral cover), then it follows that coral cover must go down in open reefs.

To demonstrate our point, we have fitted coral cover as a function of zoning status using a Beta distribution (with a logit link). As expected, the model outcomes conform our rationale and demonstrates that there is a substantial difference between open and closed reefs, with open reefs having on average ~1.8 % less coral cover (credible intervals ~1%–2.7%).

Because of the reasons mentioned above, and due to simplicity, we decided not to include this model in the revised version; however, we defer to editorial decision on whether to include this.

Minor comments:

6. Line 78- specify that this refers to the commercial catch?

Response: We changed the order of two sentences to clarify that this refers to the commercial and recreational charter catches, as follows (changes in bold):

Original: We employed a Bayesian hurdle-gamma modelling approach using paired fisheries catch and CoTS density observations encompassing three CoTS outbreak cycles (19,392 paired observations for 157 individual fisheries logbook reporting sites between 1989 and 2018, see Methods; Supplementary Methods 1-2; Supplementary Fig. 1). We focussed on six fish groups based on their reported consumption of benthic CoTS (Table 1)¹⁶ and their contribution to the commercial and recreational charter fisheries in the GBR Marine Park (Supplementary Tables 2-3)²⁵.

Revised: **Here, we focussed on six fish groups based on their reported consumption of benthic CoTS (Table 1)¹⁶ and their contribution to the commercial and recreational charter fisheries in the GBR Marine Park (Supplementary Tables 2-3)²⁵.** We employed a Bayesian hurdle-gamma modelling approach using paired fisheries catch and CoTS density observations encompassing three CoTS outbreak cycles (19,392 paired observations for 157

individual fisheries logbook reporting sites between 1989 and 2018, see Methods; Supplementary Methods 1-2; Supplementary Fig. 1).

7. *Line 101 - Please elaborate on what you mean by behavioural release of planktivorous fishes allowing CoTS larvae to reach the benthos.*

Response: We have elaborated to clarify what we mean by both ecological and behavioural release, as well as included a recommendation for further research, as follows (changes in bold):

Results

Original: This finding supports the involvement of more complex ecological effects in modulating CoTS outbreaks⁹, which have remained speculative until now, including ecological release of benthic carnivorous fishes reducing densities of invertebrates that prey on juvenile CoTS²⁷, and behavioural release of planktivorous fishes allowing CoTS larvae to reach the benthos²⁸.

Revised: This finding supports the involvement of more complex ecological effects in modulating CoTS outbreaks²⁴, which have remained speculative until now. **Such effects may include the ecological release of invertebrates that prey on juvenile CoTS, when higher numbers of large piscivores result in reduced densities of benthic carnivorous fishes²⁷. It could also include the behavioural release of planktivorous fishes away from the reef substrate with reduced numbers of large piscivores, resulting in increased number of CoTS larvae able to reach the benthos²⁸.**

Discussion

Original: Critically, our findings support the importance of hypothesised ecological and behavioural release^{37, 38}, following removal of the piscivorous coral trout, in influencing CoTS population dynamics and outbreaks.

Revised: Critically, our findings support the importance of hypothesised ecological and behavioural release^{37, 38}, following removal of the piscivorous coral trout, in influencing CoTS population dynamics and outbreaks. **The involvement of more complex ecological effects in modulating CoTS outbreaks²³ have remained speculative until now; our results warrant further research to better understand and quantify the potential role of such trophic cascades.**

8. *Line 450 - I am curious to know why the authors chose to include net and trawl fishery data, assuming there would be little interaction between CoTS / reefs and taxa caught. How do the results compare when this data is excluded from the analyses?*

Response: The commercial net and trawl fishery data were included based on the advice from QDAF that these fisheries had targeted and reported on some of the fish taxa of interest to our study in the 1990s and early 2000s. This is particularly so for coral trout which got caught and reported on by both fisheries up to 2000 (trawl fisheries) and 2004 (net fisheries), respectively (Supplementary Table 3). Hence, we believed it prudent to include the complete catch data for fish taxa of interest from both the commercial net and trawl fishery. Please note that the trawl fisheries do not report any more catches of coral reef fin fish after 2002. After 2004, the net fisheries only include very small catches of coral reef fin fish relative to the commercial line fisheries (~ <5%). To clarify, we have added the following (under Methods section 'QDAF fisheries catch data'):

“Catch data from the commercial net and trawl fisheries were included based on advice from QDAF as **both these fisheries have targeted and reported on some of these coral reef fish species.**”

9. *Line 494 - Correct LMTP*

Response: Corrected, also for another mis-spell elsewhere in the manuscript

10. *Line 515 - “...that that...”*

Response: Corrected.

11. *Line 549 - Presumably these are the “paired” reefs mentioned in the results? To provide more clarity in the methods, I suggest expanding the description of the reefs used here.*

Response: Yes, these are the paired reefs mentioned in the results. We have now included the details of the paired reef design including reefs open and closed to fishing in this section.

12. *Supp Table S4 - Are the 5 listed labrid species the only ones monitored in the LTMP? If not, why were only these five species chosen for the analyses, as opposed to all those listed as targeted by the fishery? Presumably, the main labrids caught since the ban on *Cheilinus undulatus* would be the larger tusk fish species?*

Response: Yes, these five labrid species are the only ones monitored in the LTMP. We have added the following text to clarify this (in bold):

‘All LTMP reef fish data from **all species monitored** within the families Labridae (wrasses – **five species**), Lethrinidae (emperors – **16 species**), Lutjanidae (tropical snappers – **21 species**) and Serranidae (rockcods – **35 species**) (Supplementary Table 6) [...].’

Reviewer #3 (Remarks to the Author):

The paper makes a statistical case for the "predator removal" hypothesis that "release from predation pressure is a primary cause of CoTS outbreaks," and also finds evidence of "involvement of more complex ecological effects in modulating CoTS outbreaks". It's an important finding, and the authors' suggestion that CoTS population dynamics be factored into fisheries management could be influential.

The statistical analyses involve Bayesian hierarchical modelling. It is competently done: the models follow well established procedures, the choices for distributions (including priors) are natural, and the conclusions are appropriate. Some additional analyses would be useful.

Response: Thank you for your positive comments and constructive feedback. We have addressed your comments in more detail below.

1) The authors should investigate the robustness of the conclusions to choice of prior, as this is easy to do, and it is important to understand the influence of the prior.

Response: We thank the referee for their comment. We now provide a series of figures (Supplementary Figures 5–40; 43–60; 62) which contain the direct comparison between prior and posterior for all main parameters in each model, as well as the post warm-up trace plots showing the convergence among the four chains. We can confirm that our priors were sufficiently weak and therefore have not biased the estimated posteriors.

2) It would be useful to apply the methods to simulated data to demonstrate that the methods can recover true parameters. Simulated data would also be useful to assess the impact of recreational fishing. The authors are surely correct that: "These effects are likely stronger, given that the take of emperors, tropical snappers and rockcods by the recreational (non-charter) fisheries in the GBR Marine Park is estimated to have been of similar magnitude to that of the commercial line fisheries for decades". However, it would be useful to investigate this claim using simulated data.

Response: This is a great suggestion. We have run a post-hoc power analysis by randomly drawing 100 posterior predictions generated from each model, and then re-running the original model structure on each one of the 100 draws (i.e., as if they were the original response = CoTS density). We then sampled 1,000 posterior draws for each parameter in each run and created a concatenated distribution across the 100 simulations (i.e., $100 \times 1,000 = 100,000$ draws per parameter per model). Panel a) in Supplementary Figures 5–40 demonstrates that despite some expected additional uncertainty, the range of values encompassed by the parameter estimates from the simulations are very similar to the estimates originally reported in our manuscript.

With regards to recreational fisheries, we unfortunately could not intuit exactly what the referee meant by simulated data. If we were to, e.g., double the estimates of biomass removal by assuming that recreational fisheries were of the same magnitude as commercial fisheries, we would essentially add a constant of $\ln(2)$ to the intercept in the hurdle gamma model because the link function is the natural log, and the slope estimate would not be affected. Considering a range of simulated recreational fisheries takes would certainly be a great expansion of our present study, but it is beyond the scope of what we are currently tackling.

3) The authors might want to consider applying methods of function regression to this data

set, instead of performing separate analyses for each lag x. Function regression allows the dependent and/or independent variable to be a function. Here the CoTS density as a function of lag x, and/or the biomass extraction as a function of t, could be modelled in this manner. This could improve precision if either function varies smoothly. I do not expect the authors to take up this suggestion for this paper, but they might.

Response: We appreciate the reviewer's suggestion. However, we deliberately chose to keep these as separate analyses per time lag because there is no prior knowledge that would justify a smooth transition in parameter estimates as a function of time, and in that sense our study establishes the first benchmark upon which both theoretical and experimental studies could be developed.

A few other minor comments:

1) Line 51: "a 100" should just be "100"

Response: Corrected.

2) Extended Table 1 caption should have superscript '#' on "Parrotfish".

Response: Done.

3) Line 33 of supplementary information uses label S1, all other uses exclude the 'S'.

Response: Removed the 'S' from all labels and double-checked and corrected where necessary all other labels in the Supplementary Information.

4) Line 119 of supplement: density measured in individuals per minute instead of individuals per 2 minute tow

Response: Indeed, for the purposes of CoTS modelling as a function of fish biomass removal we decided to standardise the metric to a unit minute because that was an aggregate across manta tows within a grid. The CoTS modelling in Figure 3 was conducted on the scale of individual manta tow observations (2 minutes). We have added this explanation to the Supplementary Information (pl see last sentence in Supplementary Method 4).

5) Line 144: the authors use the term 'warm-up' to describe what is more commonly referred to as 'burn-in'.

Response: Not exactly. Burn-in has been classically used in MCMC algorithms such as JAGS. However, in Stan (which uses Hamiltonian Monte Carlo), the warm-up includes an adaptation algorithm which makes it distinct from a Markov chain process. This has been explained by the Stan authors in this blog post:

<https://statmodeling.stat.columbia.edu/2017/12/15/burn-vs-warm-iterative-simulation-algorithms/>

6) Why was zoning status not included as a covariate on Line 533? A brief explanation would suffice.

Response: We believe that this explanation was already given in the preceding lines ('Fisheries in the GBR are only permitted in open reefs [...]'). Fisheries data only exist for reefs that are open to fishing, i.e., there are no data for (what would be illegal) fish removal on closed reefs, and therefore adding status (closed vs. open) is not possible for this analysis.

Reviewer comments, second review round:

Reviewer #2 (Remarks to the Author):

Thank you to the authors for their comprehensive and detailed responses to all reviewers' comments, which have further improved this solid study. I only have a few more relatively minor comments.

This study produces the most conclusive evidence of the effects of predator removal on CoTS densities so far. Although maybe necessary for the long-term survival of the Great Barrier Reef, especially in light of the increasing threats posed by climate change, the management strategies proposed here would have severe consequences for the fishing industry on the Great Barrier Reef. I would perhaps caution the authors on some of the wording especially relating to coral trout, where the links require further investigation, which is mentioned in the manuscript. Overall though, I congratulate the authors on a very interesting and relevant study!

Line 136-139: I thank the authors for adding pairwise Pearson correlation values for the different fish groups, which show varying levels of interactions. I apologise in advance if I'm missing something here, but have a further query which I didn't consider in first round of reviews. I can also only see part of the supplementary tables due to some formatting issue. Are redthroat and spangled emperors included in the Lethrinidae catch data, and similarly coral trout in the Serranidae data? If so, and glancing at Suppl. Table 3 (I am unable to see columns of the table past Serranidae), these 3 species seem to make up the majority of the catch within their respective families, then a strong correlation between these fish groups would be a given?

Line 154-156: Given you talk about the removal of coral trout in the previous and following sentences, and to make it easier for the reader to follow, I would rephrase this sentence to also reflect the effects of removal instead of higher densities of coral trout.

Line 165: I appreciate that my previous comment on there being an opposite effect at 3 years was poorly formulated. However, I still find the relatively weak, but negative effect at 3 years for Lethrinidae, Serranidae and coral trout, a little bit troubling, given the importance that is placed in the manuscript on the fish removal effects on CoTS density after 4 years. The lesser effect at 5 or 6 years can be more easily justified, if the time lag peaks at 4 years, after which the effect decreases again.

Line 209: "contribute to" instead of "drive"?

Reviewer #3 (Remarks to the Author):

The authors provide a comprehensive and compelling response to the reviews. I have no further comments or suggestions.

REVIEWERS' COMMENTS

Reviewer #2 (Remarks to the Author):

Thank you to the authors for their comprehensive and detailed responses to all reviewers' comments, which have further improved this solid study. I only have a few more relatively minor comments.

This study produces the most conclusive evidence of the effects of predator removal on CoTS densities so far. Although maybe necessary for the long-term survival of the Great Barrier Reef, especially in light of the increasing threats posed by climate change, the management strategies proposed here would have severe consequences for the fishing industry on the Great Barrier Reef. I would perhaps caution the authors on some of the wording especially relating to coral trout, where the links require further investigation, which is mentioned in the manuscript. Overall though, I congratulate the authors on a very interesting and relevant study!

Thank you.

We have taken the reviewer's comment about some of the wording around fisheries management into account. Please note, however, that the following sentences were already included in the manuscript (as acknowledged by the reviewer):

Discussion:

'The involvement of more complex ecological effects in modulating CoTS outbreaks²¹ have remained speculative until now; our results warrant further research to better understand and quantify the potential role of such trophic cascades.'

'Other less-restrictive fisheries management approaches, such as reduced fisheries take of coral reef fish species known to influence CoTS densities (Fig. 1), or temporal closures of reefs to fishing when environmental conditions conducive to outbreaks are predicted, can still contribute substantially to the recovery of reef fish biomass and associated functionality^{39,40}.'

In addition, we made the following adjustments to emphasise that to contribute more effectively to controlling and preventing CoTS outbreaks, fisheries-based management needs to be targeted to this specific purpose (changes in bold):

Abstract

'Designing **targeted** fisheries management with consideration of CoTS population dynamics offers a tangible and promising contribution to effectively reduce the detrimental impacts of CoTS outbreaks across the Indo-Pacific.'

Results, 4th paragraph, final sentence

'This indicates that **targeted** fisheries-based management of CoTS may provide an effective approach to rapidly reduce CoTS densities and contribute to preventing outbreaks across the Indo-Pacific.'

Results, final paragraph, first sentence

'These striking effects of no-take marine reserves on fish biomass (Fig. 2a), density (Fig. 2b) and length (Fig. 2c) for fish groups that influence CoTS densities within a few years (Fig. 1), portends the applicability of **targeted** fisheries-based management to prevent CoTS outbreaks.'

Discussion, 2nd paragraph:

'Furthermore, the efficacy of **targeted** fisheries-based management may well be much larger if designed with CoTS population dynamics in mind, [...]

Discussion, final sentence:

In summary, **targeted** fisheries management, including well-designed and enforced no-take marine reserves, offers a tangible and promising contribution to effectively reduce the incidence and impacts of destructive CoTS outbreaks across the Indo-Pacific region.

Line 136-139: I thank the authors for adding pairwise Pearson correlation values for the different fish groups, which show varying levels of interactions. I apologise in advance if I'm missing something here, but have a further query which I didn't consider in first round of reviews. I can also only see part of the supplementary tables due to some formatting issue. Are redthroat and spangled emperors included in the Lethrinidae catch data, and similarly coral trout in the Serranidae data? If so, and glancing at Suppl. Table 3 (I am unable to see columns of the table past Serranidae), these 3 species seem to make up the majority of the catch within their respective families, then a strong correlation between these fish groups would be a given?

The reviewer is correct. The individual species redthroat and spangled emperors are included in the Lethrinidae catch data, and similarly coral trout in the Serranidae data. We considered six fish groups, based on their reported consumption of benthic CoTS (Table 1) and their contribution to the commercial and recreational charter fisheries in the GBR Marine Park (Supplementary Table 2; Supplementary Data 1) (as presented in the 2nd sentence of the Results). Coral trout and redthroat emperor are the fish species most commonly targeted, with both redthroat and spangled emperor being well known predators of CoTS. Hence, we were interested in the potential effect of biomass removal of these species in particular, as well as the removal of all species within whole families that are targeted by fisheries (Supplementary Table 2) and are known CoTS predators (Labridae, Lethrinidae, Lutjanidae, Serranidae) (Table 1). Given that coral trout and redthroat emperor make up the majority of the catch within their respective families, we would expect a strong correlation. The important point, however, is that the correlation between coral trout and emperors is not that strong, indicating that the effect of coral trout on CoTS densities is not solely due to an interaction between the different fish groups (as discussed in the 3rd paragraph of the Results).

Line 154-156: Given you talk about the removal of coral trout in the previous and following sentences, and to make it easier for the reader to follow, I would rephrase this sentence to also reflect the effects of removal instead of higher densities of coral trout.

Good point. We have amended the sentence as follows (changes in bold):

‘Such effects may include the ecological release of invertebrates that prey on juvenile CoTS **when higher numbers of large piscivores are present; removal of such piscivores may result in increased** densities of benthic carnivorous fishes.’

Line 165: I appreciate that my previous comment on there being an opposite effect at 3 years was poorly formulated. However, I still find the relatively weak, but negative effect at 3 years for Lethrinidae, Serranidae and coral trout, a little bit troubling, given the importance that is placed in the manuscript on the fish removal effects on CoTS density after 4 years. The lesser effect at 5 or 6 years can be more easily justified, if the time lag peaks at 4 years, after which the effect decreases again.

This has indeed been puzzling, however, after re-evaluating our findings in more detail we have separated out the results for the different fish groups based on this re-evaluation.

First, the fish removal effects of *L. miniatus* and *L. nebulosus*, Lethrinidae, and Lutjanidae, were most pronounced after one and two years (Supplementary Table 1). All of these are well known CoTS predators (Table 1; Kroon et al. 2020).

Second, the fish removal effects of *Plectropomus* spp. and *Variola* spp and of Serranidae on CoTS density were most strongly manifested after four years (Supplementary Table 3). This would further support the potential involvement of complex ecological effects in modulating CoTS outbreaks.

We have addressed this in the manuscript as follows (changes in bold):

Results, 3rd paragraph, final two sentences:

‘The potential involvement of complex ecological effects is further supported by our finding that the effects of *Plectropomus* spp. and *Variola* spp and of Serranidae removal on CoTS density were most strongly manifested after four years (Supplementary Table 3). In contrast, the fish removal effects of well-known CoTS predators, namely *L. miniatus* and *L. nebulosus*, Lethrinidae, and Lutjanidae (Table 1), were most pronounced after one and two years (Supplementary Table 3).’

Results, 4th paragraph, final two sentences:

Furthermore, the fish removal effects on CoTS density were most strongly manifested within **two years for *L. miniatus* and *L. nebulosus*, Lethrinidae, and Lutjanidae, and four years for *Plectropomus* spp. and *Variola* spp, and Serranidae. This indicates** indicating that **targeted** fisheries-based management of CoTS may provide an effective approach to rapidly reduce CoTS densities and contribute to preventing outbreaks across the Indo-Pacific.

Results, 6th paragraph, 1st sentence:

These striking effects of no-take marine reserves on fish biomass (Fig. 2a), density (Fig. 2b) and length (Fig. 2c) for fish groups that influence CoTS densities within **a few** years (Fig. 1), portends the applicability of fisheries-based management to prevent CoTS outbreaks.

Line 209: “contribute to” instead of “drive”?
Changed to ‘contribute to’

Reviewer #3 (Remarks to the Author):

The authors provide a comprehensive and compelling response to the reviews. I have no further comments or suggestions.

Thank you.